

# Characterization of the basic helix–loop–helix gene family and its tissue-differential expression in response to salt stress in poplar

Kai Zhao[1], Shuxuan Li[1], Wenjing Yao[1,2], Boru Zhou[1], Renhua Li[1] and Tingbo Jiang[1]

[1] State Key Laboratory of Tree Genetics and Breeding, Northeast Forestry University, Harbin, China
[2] Northeast Institute of Geography and Agroecology, Chinese Academy of Sciences, Harbin, China

## ABSTRACT

The basic helix–loop–helix (bHLH) transcription factor gene family is one of the largest gene families and extensively involved in plant growth, development, and stress responses. However, limited studies are available on the gene family in poplar. In this study, we focused on 202 bHLH genes, exploring their DNA and protein sequences and physicochemical properties. According to their protein sequence similarities, we classified the genes into 25 groups with specific motif structures. In order to explore their expressions, we performed gene expression profiling using RNA-Seq and identified 19 genes that display tissue-differential expression patterns without treatment. Furthermore, we also performed gene expression profiling under salt stress. We found 74 differentially expressed genes (DEGs), which are responsive to the treatment. A total of 18 of the 19 genes correspond well to the DEGs. We validated the results using reverse transcription quantitative real-time PCR. This study lays the foundation for future studies on gene cloning, transgenes, and biological mechanisms.

## INTRODUCTION

The basic helix–loop–helix (bHLH) transcription factor gene family is widely existed in eukaryotes and plays an important role in plant growth and development. However, limited studies are available in plants, especially in poplar. Since the first bHLH protein structure was analyzed in 1989 (*Murre, McCaw & Baltimore, 1989*), more bHLH proteins have been identified. The bHLH gene family is named for its highly conserved bHLH domain, which consists of about 60 amino acids, including the basic region distributed at the N-terminus and the helix–loop–helix (HLH) region distributed at the C-terminus of the polypeptide chain (*Feller et al., 2011*). The basic region contains about 15 amino acids and plays an important role in target DNA recognizing and binding (*Li et al., 2006*). The HLH region, ranging from 40 to 50 amino acids in length, consists of two amphiphilic

Corresponding author
Tingbo Jiang, tbjiang@yahoo.com

alpha helices and a loop structure with uncertain length and sequence. The two amphiphilic alpha helices can form homodimer or heterodimeric to interact with other bHLH proteins (*Ellenberger et al., 1994*). The bHLH transcription factors can identify the element called E-box (5′-CANNTG-3′), with the most common one is G-box (5′-CACGTG-3′) (*Li et al., 2006*). Studies have shown that the nucleotides in the flank of the external of core element also impact specific binding (*Martínez-García, Huq & Quail, 2000*). Previous studies indicated that the known bHLH proteins in animals can be divided into six groups of A–F (*Atchley & Fitch, 1997*). In plants, however, many of the identified bHLH proteins belong to group B (*Buck & Atchley, 2003*), the most members of which are characterized by binding to G-box (*Li et al., 2006*).

The bHLH transcription factors have various functions in plant growth and development. The bHLH genes *SPT* and *ALC* impact the development of pistils (*Groszmann et al., 2010*; *Heisler et al., 2001*), and *PIL5* affects the germination of seeds in *Arabidopsis thaliana* (*Oh et al., 2004*). The bHLH gene *RGE1* in *A. thaliana* plays an important role in controlling the growth of embryos (*Kondou et al., 2008*), and the bHLH gene *SPT* is associated with final leaf size (*Ichihashi et al., 2010*). Rice bHLH gene *LAX* is the main regulator of leaf meristem (*Komatsu et al., 2003*), and rice bHLH gene *bHLH142* is involved in the development process of pollens (*Ko et al., 2014*). The bHLH genes also regulate plant response to various abiotic stresses, such as drought (*Abe et al., 1997*; *Seo et al., 2011*), salinity (*Jiang, Yang & Deyholos, 2009*; *Zhou et al., 2009*), cold (*Chinnusamy et al., 2003*; *Wang et al., 2003*), high temperature (*Koini et al., 2009*), aluminum and iron deficiency (*Kumari, Taylor & Deyholos, 2008*; *Ling et al., 2002*; *Zhang et al., 2015*), and low phosphorus stress (*Yi et al., 2005*). In addition, the bHLH genes play an important role in regulating multiple signal transduction pathways and impacting biosynthesis (*Carretero-Paulet et al., 2010*).

Poplar is an economically important tree in northern China, where salinity is one of the major abiotic stresses that limit poplar survival and growth. Since poplar is susceptible to salt, drought, cold, heavy metals, and other abiotic stresses, it is critical to identify important transcription factor gene families. In this study, we focus on 202 bHLH genes in poplar, in order to investigate their structures and functions, especially tissue-differential gene expression patterns in response to salt stress.

## MATERIALS AND METHODS

### Plant culture and stress treatment

We cut the twigs from the di-haploid *Populus simonii × Populus nigra* growing in the greenhouse. The twigs were cultivated in the same beakers with water in order to obtain new branches and roots. The conditions of culture include 60–70% relative humidity, 16-h light/8-h dark cycles, and average temperature of 25 °C. After two months, we selected 20 plants with similar growth status, which were at random divided into two equal groups with two biological replicates. One of the groups was then treated with 150 mM NaCl for 24 h. The other was used as a control. For gene expression profiling, we collected root, stem, and leaf samples from each group. The 12 samples from each group were frozen in liquid nitrogen immediately, and stored in a refrigerator at −80 °C.

## Gene expression analysis based on RNA-Seq

In order to explore tissue-differential expression patterns within the bHLH gene family in response to salinity, we sent the 12 samples to the GENEWIZ Company (https://www.genewiz.com/) for RNA-Seq using the Illumina HiSeq 2500 platform. The sequencing generated pair-end reads with 150 bases. Construction of the RNA libraries was described in our previous study (*Yao et al., 2016*). Data processing methods for sequence reads are as follows: first, we used the Trimmomatic software (*Bolger, Lohse & Usadel, 2014*) to remove the adaptor sequences and low quality sequences from the original sequencing data; second, the high quality reads were mapped to the reference genome using the STAR software (*Dobin et al., 2013*) with default parameters. Finally, gene expression levels were calculated as fragments per kilo-bases per million mapped reads applying RSEM software (*Li & Dewey, 2011*).

We identified differentially expressed genes (DEGs) by contrasting the treatment group with the control, using DESeq package (*Anders & Huber, 2010*) from Bioconductor (http://www.bioconductor.org/). We used two thresholds for DEGs selection, that is, fold change ≥2 and padj ≤0.05 (*p* value adjusted for multiple testing at the false discovery rate of 0.05).

## Phylogenetic relationship and physicochemical properties of the bHLH gene family

In order to investigate the bHLH gene family in poplar, we obtained their amino acid sequences from the PlantTFDB database (*Jin et al., 2017*). Then the multiple sequence alignment was carried out by use of ClustalX 1.83, and the phylogenetic trees were constructed with MEGA 5.05 (*Tamura et al., 2011*; *Thompson et al., 1997*). The parameters are as follows: the neighbor-joining (NJ) method, 1,000 iterations of bootstrap resample and Poisson model. Since we are interested in general sequences similarities, we then classified the proteins into grand groups. We used bootstrap resample to support the classification. A threshold of the bootstrap score 50 is applied, which means 50% of the chances proteins can be found in the same group.

In addition, we constructed an evolutionary tree, using the best model selected by the SMS software and maximum likelihood (ML) method (*Lefort, Longueville & Gascuel, 2017*). We obtained the DNA binding domain sequences of poplar bHLH family from the PlantTFDB database, Pfam database, and SMART (*Finn et al., 2016*; *Jin et al., 2017*; *Letunic & Bork, 2018*; *Letunic, Doerks & Bork, 2015*), followed by performing multiple sequence alignment. The DNA binding domain obtained can be visualized by WebLogo (*Crooks et al., 2004*; *Schneider & Stephens, 1990*). The physicochemical properties of proteins, including length, molecular weight, theoretical isoelectric point, aliphatic index, and grand average of hydropathicity, were predicted by ProtParam (*Gasteiger et al., 2005*).

## Protein sequence motif analysis

Protein sequence motifs were identified by the MEME method (*Bailey et al., 2009*), and then the motif logos were obtained using the TBtools (https://github.com/CJ-Chen/TBtools).

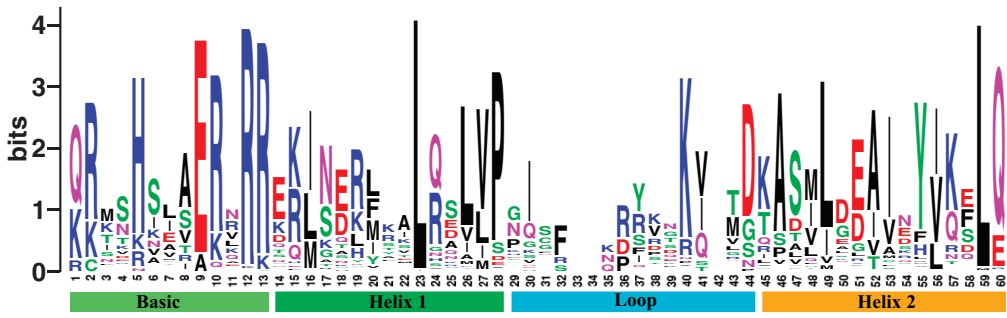

**Figure 1 The DNA binding domain alignment logo of poplar bHLH family.**

Annotations of motifs were obtained from Pfam database and InterProScan (*Finn et al., 2016*, *2017*; *Jones et al., 2014*).

## Gene structure analysis

We also downloaded the gene sequences from Phytozome (*Goodstein et al., 2012*; *Tuskan et al., 2006*). Structures of the bHLH gene family members were derived using Gene structure Display Server Program (*Hu et al., 2015*).

## Subcellular location and gene coordinate analysis

Subcellular localization of bHLH proteins was predicted by WoLF PSORT (*Horton et al., 2007*). Gene coordinate was identified by PopGenIE v3.0 database (*Sjodin et al., 2009*).

## Verification by reverse transcription quantitative real-time PCR

In order to validate our results from RNA-Seq, we performed reverse transcription quantitative real-time PCR (RT-qPCR) with 30 DEGs selected at random. Actin is used as a reference gene (*Regier & Frey, 2010*). The primer sequences are listed in Table S1. The experimental procedures were described in our previous study (*Yao et al., 2016*). The expression level of each gene was calculated as relative to corresponding gene expression in leaf with no treatment.

# RESULTS

## Identification of bHLH gene family and analysis of its physicochemical properties

We obtained 202 bHLH gene members in poplar from the PlantTFDB 4.0 (*Jin et al., 2017*). The gene family shares the DNA binding domain. We named them *PtrbHLH1* to *PtrbHLH202* (Table S2). The aligned DNA binding domain sequences of poplar bHLH family are showed in Table S3, and the DNA binding domain alignment logo with the length of 60 amino acids is displayed in Fig. 1. Evidence from physicochemical properties indicted that lengths and molecular weights of the 202 corresponding proteins vary substantially (Table S2). The average length is 339.4 amino acids, ranging from 60 to 742. The mean of molecular weights is 37,634.23 Da (6,917.34–79,784.41 Da). The theoretical isoelectric points of these proteins are between 4.63 and 9.92. The aliphatic indexes are in
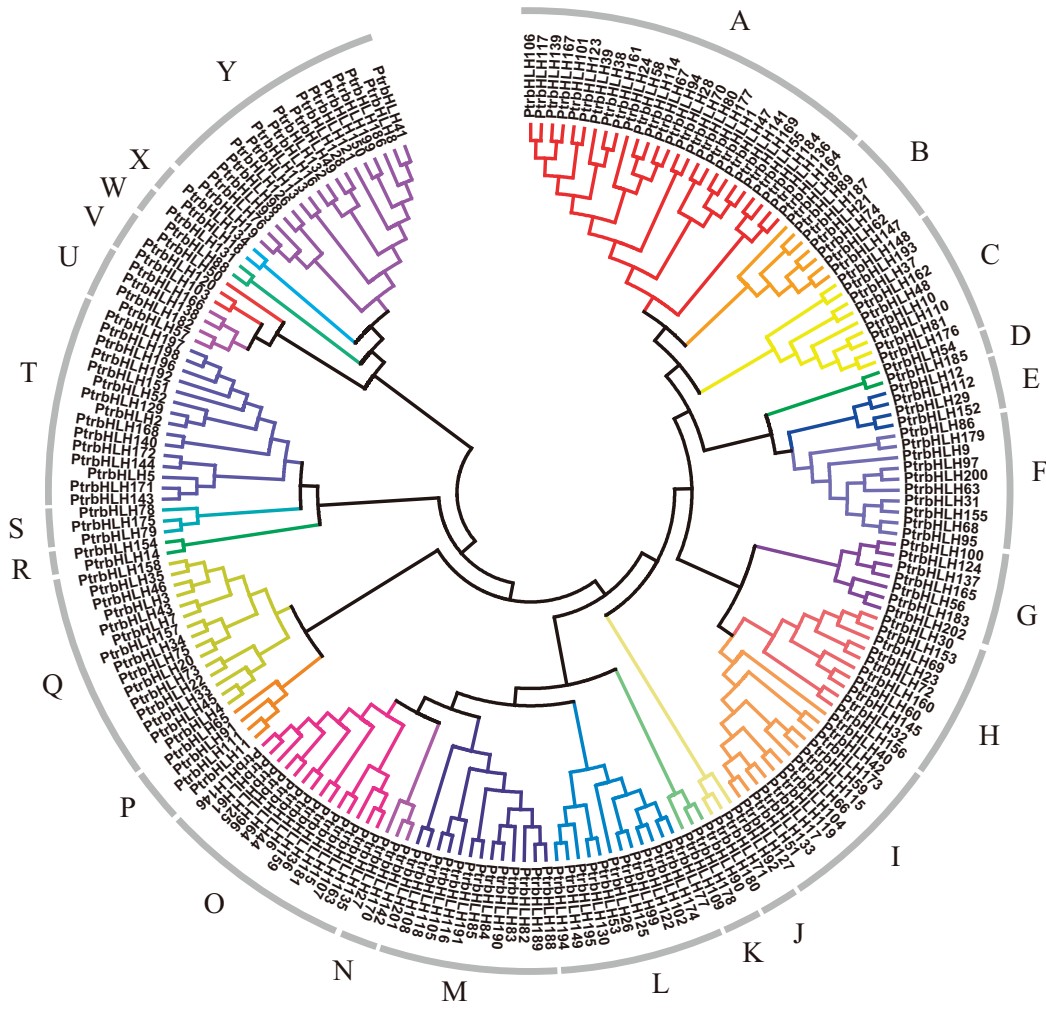

**Figure 2 Dendrogram of bHLH gene family proteins.** Each color represents a special group.

the range of 32.67–108.33, which indicate that thermal stability of the proteins varies substantially. The grand average of hydropathicity runs the gamut of −1.885 and −0.06, indicating that these proteins are hydrophilic.

## Phylogenetic trees and protein sequence motif analysis

In order to explore relationships of the bHLH gene family members in poplar, we constructed a dendrogram and a NJ-phylogenetic tree with their protein sequences (Fig. 2; Fig. S1). The proteins can be divided into 25 groups (A–Y). The reliability of the results is supported by the bootstrap resample test (Fig. S1). The average size of the groups has approximately eight members, ranging from 2 to 26. To validate our phylogenetic tree, we also constructed a ML-phylogenetic tree with the best model of JTT + G + F (Fig. S2). All of groups have a good consistence on the ML-phylogenetic tree except group J and V with only six members.

As expected, members in each group, especially closely related proteins, have the same or similar conserved motifs. We obtained 15 conserved protein sequence motifs based on

the MEME (*Bailey et al., 2009*). It is shown that motifs 1, 2, 3, 4, 6, 7 were annotated to the bHLH domain, by use of Pfam and InterProScan (*Finn et al., 2016*, *2017*; *Jones et al., 2014*). Motifs 2 and 4 share the E-box/N-box specific site. Motif 5 is the ACT domain. Motif 8 is achaete-scute transcription factor-related. Motif 9 is the bHLH-MYC and R2R3-MYB transcription factors N-terminal. The others have no annotation (Table S4).

Based on MEME (*Bailey et al., 2009*), all the bHLH family genes contain the motif which was annotated as the bHLH domain, except *bHLH202* without any motif (Fig. S3). Group A, B, C, E mainly contain motifs 1 and 3; 13 groups largely share motif 2; group O harbors motif 4; 7 groups mainly have motifs 6 and 7; and several proteins in multiple groups contain motifs 1, 2, or 3. Regarding other motifs, motif 5 is mainly distributed in group K–R; motif 8 in group T; motifs 9, 11, 12, 13 in group Q; motif 10 in group B and Y; motif 14 in group I; motif 15 in group K, N, O, P, Q. In addition, group F only contains motif 2, group U only harbors motifs 6 and 7, and group Y only has motifs 2 and 10. Motif 4 only occurs in group O.

## Gene structure at the DNA level

In order to compare similarities of the gene structure in each group, we characterized introns and exons of each gene (Fig. S3). In general, the 202 genes differ substantially by length and the number of introns and exons. The majority of the bHLH gene family members have multiple introns, except for a few genes without a single intron (Fig. S3).

Within each of the 25 protein groups shown in the Fig. 2, multiple subgroups occur based on gene structure. For example, the first four members in group A share the similar number of introns and exons, as well as the length of the exons. Similar situation is observed for the last four members in group C (Fig. S3).

## Subcellular localization and gene coordinate

In order to characterize cellular distribution of the bHLH proteins, we predicted the localization of these proteins using the WoLF PSORT procedure (*Horton et al., 2007*). The results show that the 181 bHLH proteins were predicted to locate in the nucleus, six proteins in the cytoplasm, 10 proteins in the chloroplast, three proteins in the mitochondria, and one protein in the Golgi apparatus. In addition, one protein was predicted to be double localization with shuttling between the cytoplasm and the nucleus (Table S5).

In order to examine chromosomal distribution of the bHLH gene family, we applied the PopGenIE v3.0 database (*Sjodin et al., 2009*). The results indicate that the bHLH gene family members are unevenly distributed on all 19 chromosomes. Chromosome 2 harbors a maximum number of 22 genes; in contrast, each of chromosomes 16, 17, and 18 contains a minimum of only six genes (Table S5). And the distribution is independent on the size of the chromosome. For example, chromosomes 1, the largest one, harbors 18 genes. Chromosomes 9, the minimal one, contains nine genes.

## Tissue-differential gene expression without salt treatment

The sequencing results are given in Table S6. The sequencing depths are approximately tenfold. Total reads for each sample range from 31.8 to 38.3 million. The total mapped

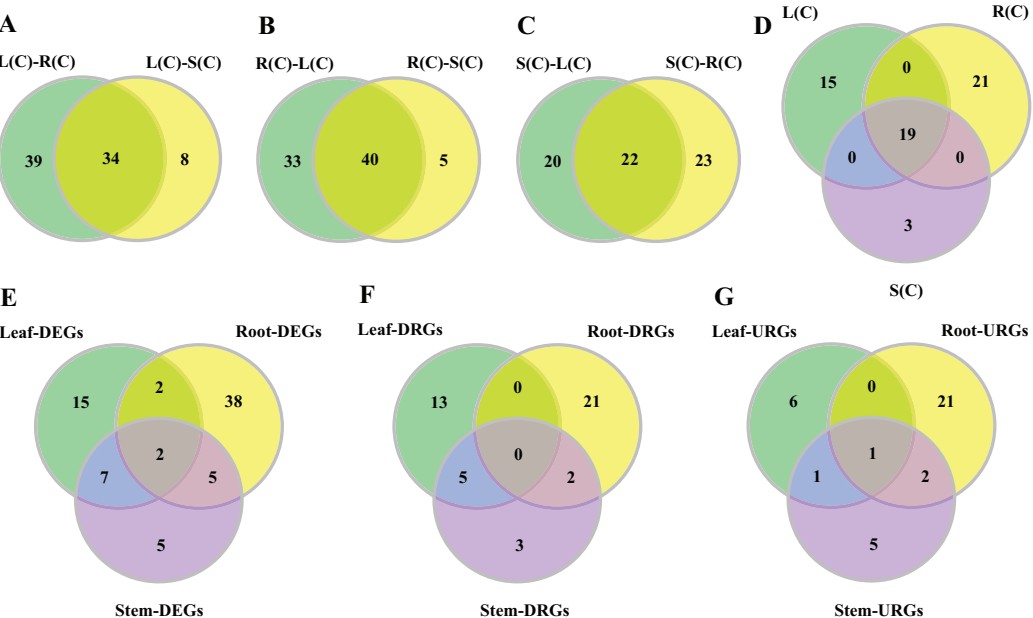

**Figure 3** **Venn diagrams of tissue-differential bHLH genes and DEGs challenged with salinity.** (A–C) Number of genes displaying distinct and shared expression in tissue pairs without salt treatment. The shared genes express differently in two tissue pairs. (A) Comparison between leaf–root and leaf–stem pairs. (B) Comparison between root–leaf and root–stem pairs. (C) Comparison between stem–leaf and stem–root pairs. (D) Comparison between the three shared parts. We extracted the shared genes from each of the comparisons mentioned above, and compared them. (E–G) Number of DEGs, DRGs, or URGs in response to salt stress in each tissue.

reads run the gamut of 27.3 (77.3054%) and 36.4 (95.044%) million. The Pearson correlation coefficients of uniquely mapped reads between biological replicates are from 0.961 to 0.994, which are highly significant ($p$-values $< 2.2e-16$).

To explore tissue-differential expression of the gene family without salt treatment, we first compared three sets of the tissue pairs, that is, leaf–root, leaf–stem, and root–stem (Data S1). Then we contrasted between the tissue pairs and retrieved genes that are shared. Results are shown in Figs. 3A–3D.

In the comparison of the leaf–root pair, we identified 73 genes whose expression levels differ significantly. Similarly, we found 45 and 42 genes in the root–stem and leave–stem pairs, respectively (Data S1). We then focused on inter-pair comparisons. Regarding the root–leaf and root–stem pairs, 40 genes are shared (Fig. 3B), indicating the gene expressions are different in roots relative to both leaves and stems. Similarly, 34 genes are shared in the leaf–root and leaf–stem pairs, followed by 22 genes in the stem–leaf and stem–root pairs (Figs. 3A and 3C). Among the three sets of shared parts, 19 genes are shared (Fig. 3D; Table S7).

A heatmap regarding the 19 gene expressions across the three tissues is given in Fig. 4. As expected, there are distinct gene expression patterns for the 19 genes. In general, the 19 genes can be classified into three clusters that are tissue differential. Cluster 1 genes are highly expressed in leaves, lowly expressed in roots, and moderately expressed in stems. Cluster 2 genes are greatly expressed in stems, and lowly expressed in roots and leaves.

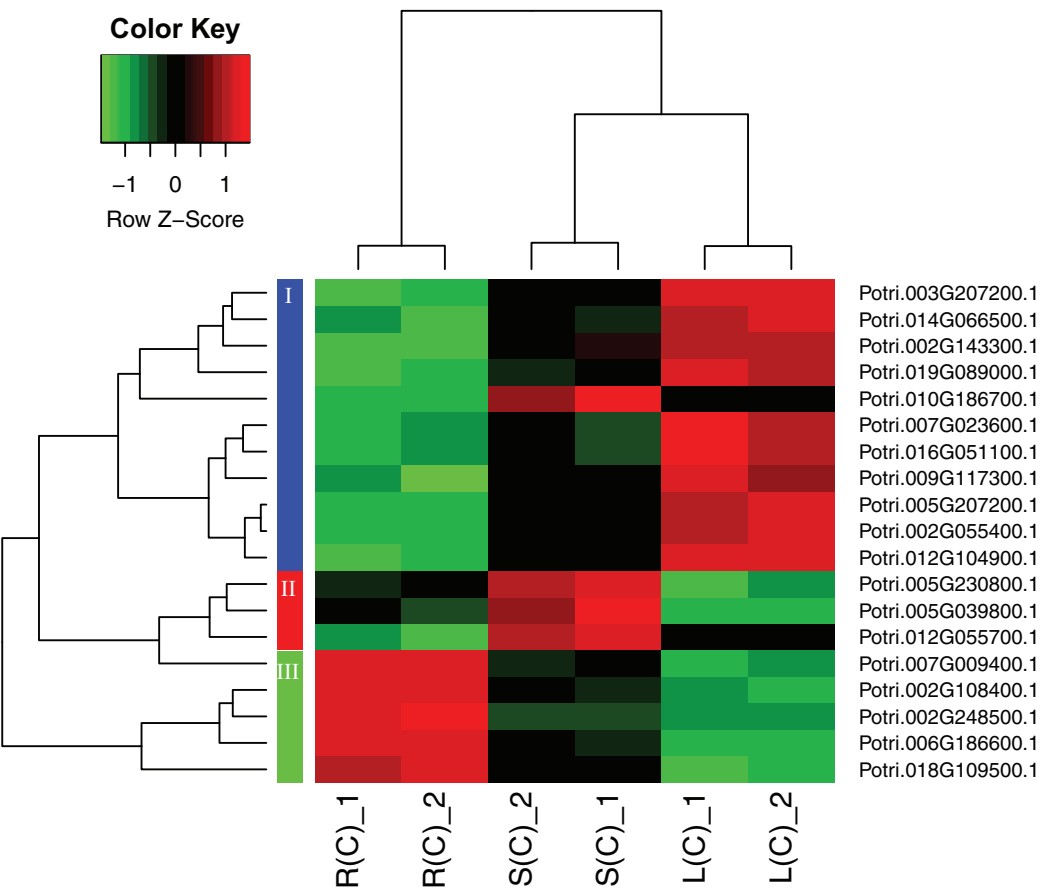

**Figure 4 A heatmap of genes displaying tissue-differential expression without treatment.** We focused on the 19 genes that have tissue-differential expression. The gene expression values are square-root transformed fragments per kilo-bases per million mapped reads (FPKM). We used the *Z*-score as standardization method for each gene. We used dist (distance matrix computation) function with Euclidean method to compute and return the distance matrix and hclust (hierarchical clustering) function with complete method to perform a hierarchical cluster analysis. The colorful vertical bars on the left side denote gene cluster 1–3 orderly.

Cluster 3 genes are highly, moderately, and lowly expressed in roots, stems, and leaves. Cluster 1 genes display an opposite pattern to cluster 3 genes.

In order to explore the direction of differential expression in different comparisons, we classified the tissue-differential genes into up- and down-regulated ones (Fig. S4). We found that most of the tissue-differential genes in leaves have similar trends, for instance, there are respective 15 genes consistently down-regulated and 17 genes agreeably up-regulated in comparison between the leaf–root and leaf–stem tissue pairs. In contrast, only two genes display inconsistency. Similar results are observed in the comparison between the root–leaf and root–stem tissue pairs. However, there is no such trend in the stems.

## Differential expression genes in response to salinity

Screen for salinity responsive members in the bHLH gene family is critical for mechanistic understanding of gene regulation of stress tolerance. In the leaves treated with salt stress,

we identified 26 DEGs in the family. A total of 8 of the genes are up-regulated and 18 are down-regulated in treatment samples compared to controls. In roots, 47 DEGs were found, with 24 up-regulated and 23 down-regulated. In stems, 19 DEGs were obtained, with nine up-regulated and 10 down-regulated (Data S2). The number of up-regulated DEGs (URGs) in roots is larger than the down-regulated DEGs (DRGs), which is opposite to the DEGs in leaves or stems. Among the URGs, variation of gene expression is remarkably greater in roots compared to that in the other two tissues. In contrast, among the DRGs, leaf tissue displays the widest gene expression variation (Fig. S5).

The expression degrees of the DEGs in each tissue are shown in Fig. S6. The majority of the DEGs are concentrated at two- to eightfold changes. It is only in roots that six genes with over-expressing of >16 fold changes. In contrast, 11 genes that are down-regulated with >16 fold changes are mainly in leaf tissue.

## Tissue-differential gene expression in response to salinity

In order to observe the distribution of DEGs in different tissues, we drew the Venn diagram (Fig. 3E). There are 74 DEGs that are responsive to salt stress in at least one tissue. There exist 16 (21.6%) of the DEGs in any two of the three tissue combinations, with nine (12.2%) in the leaf–stem pair, seven (9.5%) in the root–stem pair, and four (5.4%) in the leaf–root pair. Only two (2.7%) DEGs are shared across the three tissues. There are 38 (51.4%) DEGs that are specifically occurred in roots, followed by 15 (20.3%) in leaves, and 5 (6.8%) in stems (Fig. 3E). The gene list is given in Table S8. Number of respective DRGs and URGs are shown in Figs. 3F and 3G. Across the three tissues, the majority of DRGs and URGs are in the root tissue.

Two heatmap plots regarding the expression of 74 DEGs across the three tissues are displayed in Fig. 5. In general, the URGs, which are classified into four clusters, display distinct patterns that contrast between the tissues (Fig. 5A). For example, the top four genes in cluster 1 are highly expressed in roots and stems, but lowly expressed in leaves. Conversely, all the genes in cluster 3 display over-expression in roots, but low expression in other two tissues. The DRGs, which are classified into four clusters, exhibit similar patterns (Fig. 5B). For example, all the genes in cluster 1 are highly expressed in leaves and stems, but lowly expressed in roots. In contrast, cluster 4 genes display an opposite pattern compared to the cluster 1 genes.

To connect the tissue-differential genes without salt treatment with the DEGs in response to salinity, we mapped the 19 genes onto the heatmaps of the DEGs (Fig. 5). In general, there is a good correspondence between the matched genes, regarding tissue difference. For example, the cluster 1 genes from Fig. 4 and highlighted in blue in Fig. 5 are highly or moderately expressed in leaves and stems.

## Validation of the DEGs by RT-qPCR

In order to validate the DEGs in response to salt stress, which are identified by RNA-Seq, we selected at random 30 genes for RT-qPCR. Results turn out to be that the trend of the relative gene expression is comparable to that from RNA-Seq (Fig. 6). For example, the salt-responsive DEGs in leaf, such as *Potri.002G054100.1* and *Potri.002G248500.1*, are

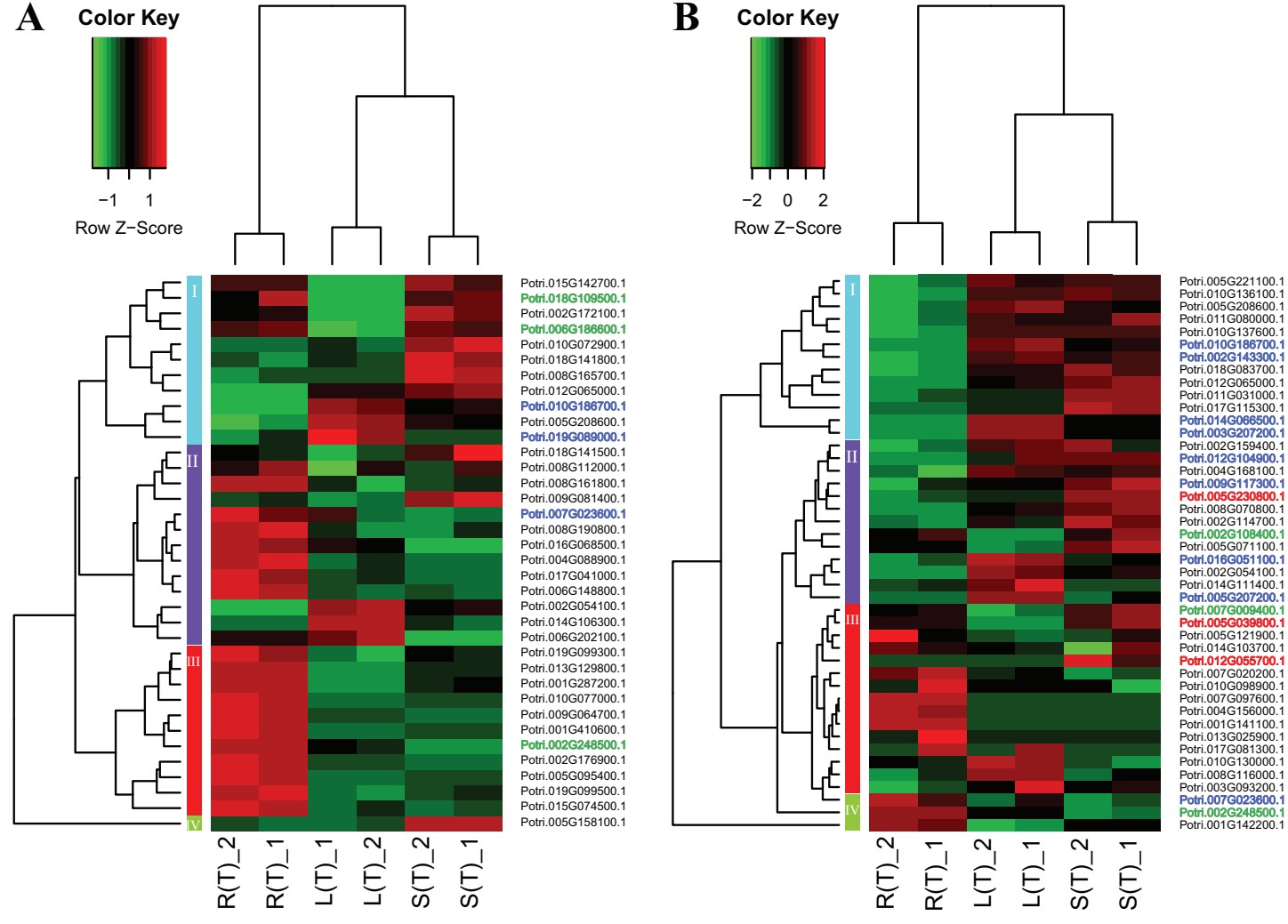

**Figure 5 Two heatmaps of URGs (A) and DRGs (B).** The gene expression values are square-root transformed fragments per kilo-bases per million mapped reads (FPKM). We used the *Z*-score as standardization method for each gene. We used dist (distance matrix computation) function with Euclidean method to compute and return the distance matrix and hclust (hierarchical clustering) function with complete method to perform a hierarchical cluster analysis. The colorful vertical bars on the left side denote gene cluster 1–4 orderly. The genes, which are highlighted in blue, red, and green, belong to the cluster 1–3 in Fig. 4 orderly.

up-regulated with over two fold changes in both RNA-Seq and RT-qPCR. In contrast, DEGs in leaf, such as *Potri.012G055700.1* and *Potri.009G117300.1*, are down-regulated with over eight fold changes.

Congruent results are observed for DEGs in stems. For example, the salt-responsive DEGs in stems, such as *Potri.005G158100.1*, *Potri.002G172100.1*, and *Potri.008G165700.1*, are over-expressed with over two fold changes in both experiments. *Potri.012G055700.1* is down-regulated with over seven fold changes.

Similar results are obtained for DEGs in roots. For example, the salt-responsive DEGs in roots, such as *Potri.012G104900.1* and *Potri.004G156000.1*, are up-regulated over two fold changes in both experiments. In contrast, *Potri.002G054100.1* is down-regulated over 3.5 fold changes.

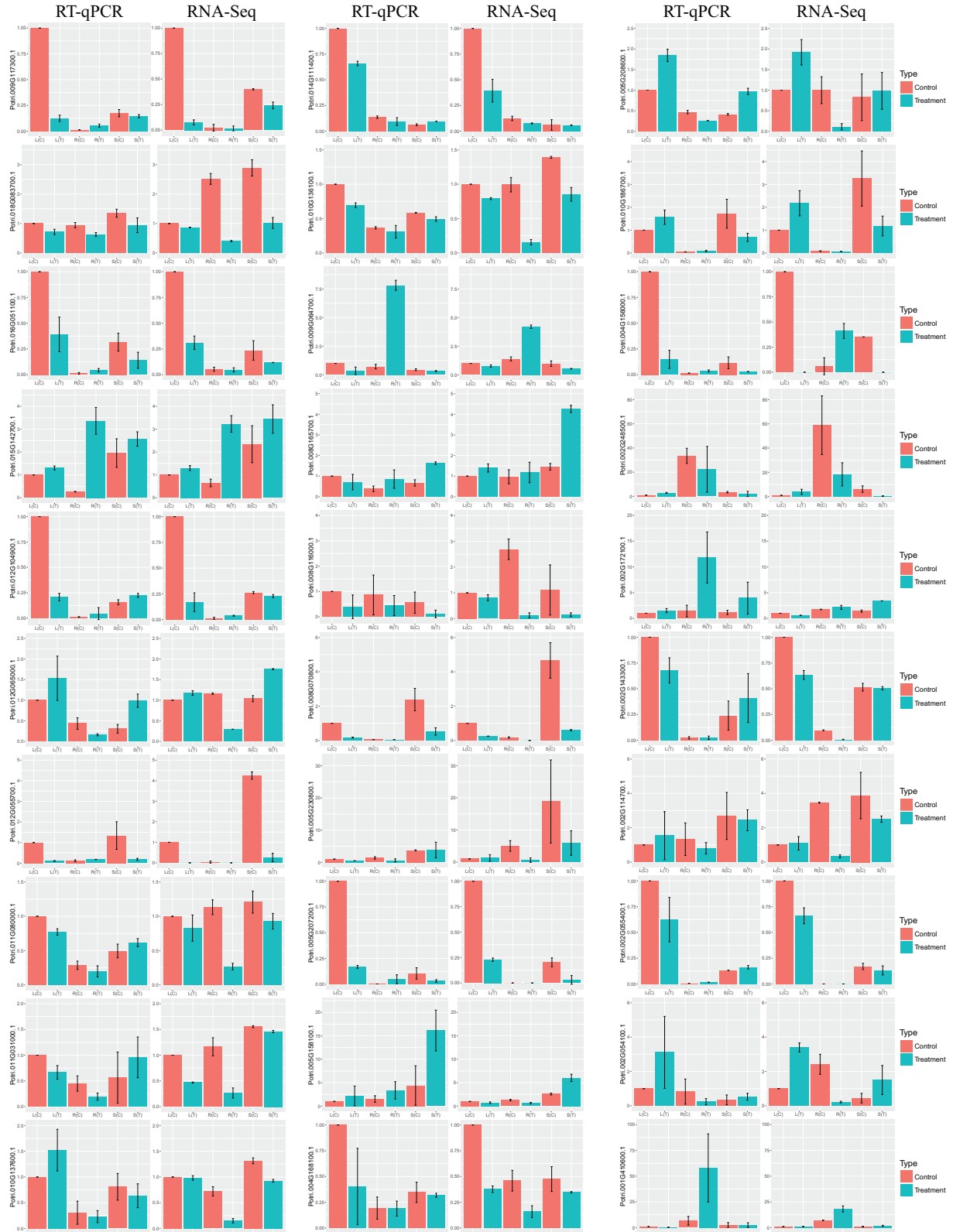

**Figure 6 Barplots of DEGs relative expression levels based on RT-qPCR and RNA-Seq.** The expression levels of each gene were calculated as relative to corresponding gene expression in leaf with no treatment. Error bars represent SD of biologic replicates.

## DISCUSSION

The bHLH gene family is one of the largest families in plants. In previous studies, the bHLH genes in Arabidopsis, poplar, rice, moss, and algae had been updated and classified in 2010, and found 183 bHLH genes in *Populus trichocarpa* using Joint Genome Institute (JGI) Ptri version 1.1 (*Carretero-Paulet et al., 2010*). However, part of the gene IDs used in the previous study are too inconvenient to be found in the updated database. The gene annotations in the updated database are also more complete. In this study, we obtained 202 poplar bHLH genes from the PlantTFDB database which identifies transcription factors applying JGI Ptri version 3.0 (*Jin et al., 2017*). We have also characterized the gene and protein structures of the 202 bHLH gene family members. According to the bootstrap values in the NJ-phylogenetic tree, the gene family can be classified into 25 distinct groups. The groups correspond well to the ML-phylogenetic tree. Each group harbors specific sequence motifs that are corresponding to different protein domains. In addition, each group shares similar intron and exon structures. These results of motif and gene structure analysis are similar to the study of the poplar trihelix family (*Wang et al., 2016*). The previous study found that the domains outside the bHLH domain may have diverse molecular functions in Arabidopsis (*Toledo-Ortiz, Huq & Quail, 2003*). For example, the PAS domain of PIF3 plays a major role in the interaction with phytochrome B (*Zhu et al., 2000*). The poplar bHLH transcription factors also contain many different motifs outside the bHLH domain, which suggests that poplar bHLH gene family members have different functions. Since the functional studies of bHLH transcription factors are limited in poplar and there are many amino acid motifs whose functions are unknown in this family, the functional diversities of poplar bHLH transcription factors need to be explored in the further study.

For the subcellular localization prediction, most of the bHLH proteins were predicted to locate in the nucleus except for 21 members locating in other organelles. We speculate that they may play a role in these organelles and enter the nucleus by interacting with other transcription factors or under stress conditions. Also, we found that nine of the 21 members inducible by salt stress.

Due to lack of studies on poplar regarding the gene family, we focused on investigating tissue-differential gene expression. We have identified 19 genes that demonstrate tissue-differential gene expression patterns. The 19 genes can be classified into three distinct clusters.

To better understand functions of the 19 genes, we then annotated the genes onto the genome of *A. thaliana* using Phytozome (Table S9) (*Goodstein et al., 2012*; *Lamesch et al., 2012*). The best-hit homologous gene of *Potri.002G108400.1* is *AT5G65640*, which promotes flowering under short-day condition (*Sharma et al., 2016*). *AT2G24260*, the homologous gene of *Potri.006G186600.1*, plays a role in root development (*Lin et al., 2015*). *AT1G73830*, the homologous gene of *Potri.012G055700.1*, is required for normal growth of *A. thaliana* (*Friedrichsen et al., 2002*). *AT2G18300*, the homologous gene of *Potri.007G023600.1*, is associated with *A. thaliana* growth and immune antagonism (*Malinovsky et al., 2014*). These studies suggest that the genes are important in the growth and development of poplar.

Based on the analysis of the expression levels of bHLH genes under salt stress, we have identified 74 DEGs that are responsive to the stress in at least one tissue. There exist 16 (21.6%) DEGs in the three tissue pairs, with four (5.4%) DEGs in the leaf–root pair, nine (12.2%) in the leaf–stem pair, and seven (9.5%) in the root–stem pair. Only two (2.7%) DEGs are shared across the three tissues, suggesting that different tissues may have diversified mechanisms in the regulation of response of salinity in poplar. There are 38 (51.4%) DEGs that are specifically occurred in roots, followed by 15 (20.3%) in leaves and five (6.8%) in stems (Fig. 3E). It is clear that the root and leaf tissues play a more significant role in salt stress responses, compared to stem tissue. The enrichment of DEGs in root is associated with the fact that root is a primary stress perception and response organ.

By annotating the sequences of the DEGs onto the genome of *A. thaliana*, we found that homologous genes in Arabidopsis are associated with plant growth, development, and stress response (Table S9) (*Goodstein et al., 2012*; *Lamesch et al., 2012*). For example, *AT2G18300*, the best-hit homologous gene of *Potri.005G121900.1*, is related to the growth and immunity of Arabidopsis (*Malinovsky et al., 2014*). *AT1G59640*, the best-hit homologous gene of *Potri.008G190800.1*, plays a part in Arabidopsis petal growth (*Varaud et al., 2011*). *AT3G47640*, the best-hit homologous gene of *Potri.015G142700.1*, regulates response to iron deficiency in Arabidopsis roots (*Long et al., 2010*). *AT1G51140*, the best-hit homologous gene of *Potri.003G207200.1*, impacts stomatal opening (*Takahashi et al., 2013*). In this study, the *Potri.003G207200.1* gene was down-regulated with >32 fold changes in poplar roots challenged with salinity, compared to the control (Data S2). Since the stomata plays an important role in regulating water balance under stress, we speculate that *Potri.003G207200.1* gene expresses in roots may regulate signal transduction pathways that are related to stomatal opening and closing in response to the stress.

Then, we mapped these genes into the gene family groups, and found that the genes, which show tissue-differential expression patterns, distribute in nine groups unequally, and salt stress response genes widely distribute in 20 groups. It should be noted that many gene pairs among these genes are in the closest position of the phylogenetic tree. It also demonstrates that the genes, which have closer genetic relationship, may have similar functions.

It is interesting to compare the genes that have tissue-differential expression before and after salt treatment. Before the treatment, we identified 19 significant genes, with 11 over-expressed in leaf, followed by five in roots and three in stems (Fig. 4). After the treatment, we found 74 significant DEGs in response to salinity (Fig. 3E). The DEGs specific to root, leaf, and stem are 38, 15, and 5, respectively (Fig. 3E).

It is noteworthy that the 18 of the 19 bHLH genes showing tissue-differential expression without treatment correspond well to those DEGs that are responsive to salinity (Fig. 5). We then selected at random 30 DEGs for RT-qPCR for validation. The results from RNA-Seq and RT-qPCR are congruent (Fig. 6).

## CONCLUSION

In this study, we focus on the 202 bHLH gene family members in poplar, starting from analysis of their physicochemical properties, evolutionary relationship, and gene

structures, followed by tissue-differential gene expression and differential expression in response to salinity. Around one-third of the genes are found to play a significant role in regulating salinity response. The majority of the DEGs display gene expression patterns in a tissue-differential fashion. This study lays the foundation for future work in gene cloning, transgenes, and biological mechanisms.

### Funding
This work was supported by the National Natural Science Foundation of China (31570659) and the 111 Project (B16010). The funders had no role in study design, data collection and analysis, decision to publish, or preparation of the manuscript.

### Grant Disclosures
The following grant information was disclosed by the authors:
National Natural Science Foundation of China: 31570659.
111 Project: B16010.

### Competing Interests
The authors declare that they have no competing interests.

### Author Contributions
- Kai Zhao performed the experiments, analyzed the data, contributed reagents/materials/analysis tools, prepared figures and/or tables, authored or reviewed drafts of the paper, approved the final draft.
- Shuxuan Li analyzed the data, authored or reviewed drafts of the paper, approved the final draft.
- Wenjing Yao analyzed the data, authored or reviewed drafts of the paper, approved the final draft.
- Boru Zhou conceived and designed the experiments, authored or reviewed drafts of the paper, approved the final draft.
- Renhua Li prepared figures and/or tables, authored or reviewed drafts of the paper, approved the final draft.
- Tingbo Jiang conceived and designed the experiments, contributed reagents/materials/analysis tools, authored or reviewed drafts of the paper, approved the final draft.

### Data Availability
The raw data are provided in the Supplemental Dataset Files.

### Supplemental Information
Supplemental information for this article can be found online at http://dx.doi.org/10.7717/peerj.4502#supplemental-information.

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
