# Peer review of "Characterization of the basic helix–loop–helix gene family and its tissue-differential expression in response to salt stress in poplar"

_PeerJ, doi:10.7717/peerj.4502_

## Round 0.1 · original submission · Major Revisions

All the four reviewers think the topic is interesting and the compiled data and analysis is useful for the bHLH family TF study. Meanwhile they also proposed essential revision request regarding to the manuscript. I suggest the authors to response those concerns and improve the manuscript.

In addition, Reviewer 2 provide the detailed report as follows,

This study explored the genetic and physicochemical properties of bHLH genes. Transcriptomic profiling using RNA-Seq identified tissue-specific pattern and some of the results were validated by qPCR. The analysis conducted in this study is comprehensive and got some interesting result. 1, However, my major concern is that the statistical analysis is under-power considering the differential expression is based on only two replicates. This makes the result less convincing to me.
2, Another comment regarding the analysis is for subsection “Tissue-differential gene expression without salt treatment”. The authors should classify the tissue-differential genes into up-regulated and down-regulated since the direction of differential expression in different comparisons may be different.
3, In addition, the subsection “Validation of the DEGs by RT-qPCR” is not informative. The authors need add detailed and quantified description of the similarity between RNAseq and qPCR.
4, Finally English language need to be improved for more clear understanding. For examples, in the abstract, “before salt treatment” should be “under salt treatment”; “between-pair comparisons” should be changed to “inter-pair comparisons”; change subtitle “Verified by quantitative real-time PCR” to “verification by quantitative real-time PCR”, …

Reviewer 1 ·

Basic reporting

no comment

Experimental design

no comment

Validity of the findings

no comment

Additional comments

In this submitted manuscript, Zhao et al. analyze the salt stress response of bHLH family TFs on poplar. The bHLH family TFs are not only crucial for the embryonic development of many fundamental organ systems but also important for cell reprograming, i.e. Ascl1, NeuroD1 and Myod1. They are very flexible in terms of regulating transcription in that they can either promote or repress transcription, and do so in many different ways. The manuscript is technically solid. I have some minor comments.
1, There are some concepts don’t have clear definition, i.e. bootstrap values. Authors should give some interpretation before using these concepts. My suggestion is modifying the sentence in abstract “According to their protein sequence similarities and the bootstrap values, the genes can be classified into 25 groups with specific motif structures each.” as “According to their protein sequence similarities, we classified the genes into 25 groups with specific motif structures.”
2, Please explain why the number of samples in raw 65 (we selected 20 plants) is not consistent with the number in raw 74 (profiling from the 12 samples…).
3, Why don’t use the gene symbol in Figure 1? I have no idea on PtrbHLH41 and PtrbHLH41 are similar on protein sequence.
4, Authors should give some description on which clustering method and which distance matrix are used in gene expression clustering in Fig. 4 and Fig. 5.
5, The raw data in supplemental files is not the whole data. Authors should upload the whole gene expression data of 202 genes on 12 sample, not only the differential expressed genes’ data.
6, The author should improve the readability.

Reviewer 2 ·

Basic reporting

The English in this manuscript need to be improved.

Experimental design

no comment

Validity of the findings

With only two biological replicates, the statistical analysis is under power. This should be stated in the analysis.

Additional comments

The author should expand subsection "Validation of the DEGs by RT-qPCR" by comparing RT-qPCR and RNA-seq result in more detailed and quantified way.

Reviewer 3 ·

Basic reporting

The manuscript is written well, except minor grammatical mistakes e.g. at line 24.

Experimental design

Analyses are relevant in the context of bHLH transcription factor family. However, most of the manuscript is descriptive without much discussion of mechanistic basis of the biology of bHLH family genes.

Validity of the findings

The findings, particularly RNA-seq data may be useful for the researchers interested in bHLH transcription factor in poplar.

Additional comments

- Authors should include a representative structure of the consensus bHLH domain.

- The grouping of bHLH in 25 groups seem to be arbitrary (Fig1, S1). It is unclear what constitutes a group, and in many cases group divisions seem random. For example, Groups W, X, Y can be classified as a single group. And group A can easily be divided into 3 groups or more.
Authors should make the criteria to define a group absolutely clear and discuss the unique properties of each group in detail. Also, there does not seem to be a pattern in Fig 2 in gene structure and motifs belonging to different groups.

- Is the phylogeny constructed by fitting the best model. It is unclear how the optimal parameters for phylogeny were obtained. Authors should evaluate best parameters using a model selection tool.

- Fig 1 is better described as dendrogram and not phylogenetic tree, because there is no information about distance measure and branch lengths in this tree.

- Subcellular location prediction is interesting. However, authors should discuss more about why a transcription factor will be located to Golgi? And if there is any difference in function, motifs, gene structure etc among the bHLH located in different cellular compartments.

- Authors should discuss the source of functional diversity of HLH transcription factors. It seems that the
functional diversity would be attributed primarily to domains and motifs other than HLH domain. But
nothing has been discussed on this front in the paper. The implications in the diversity of HLH motifs are also unclear.

- What are the other up/down regulated genes in stress RNA-seq? Authors should discuss how the target genes of bHLH are affected by stress.

- RNA-Seq data should be deposited in the public databases and accession numbers should be included in the data accessibility section.

- It will be informative to include the phylogenetic groups for the motifs in Table S3.

Reviewer 4 ·

Basic reporting

The paper focuses on studying bHLH class of genes in poplar plant.

The language in the paper needs clarification or grammatical correction. Specifically,
- line 19: "Besides" should not be used
- line 84: "standards" is actually referring to thresholds applied to filter the gene list

Legends of the figures need to be further clarified. Specifically, figure 3 description is confusing with reference to sub panel A-D more than once. Figure 5 legend needs to clarify which heatmap corresponds to URG and which one to DRG.

The RNA-Seq data should be uploaded in SRA and the SRA ID specified in the paper.

Experimental design

The Abstract needs to clearly define the objective/aim of the study.

The authors seem to set out to study the bHLH gene family however it is not well defined on why the author choose salinity to study genomic responses under abiotic stress.

Methods need some clarifications:
- Line 71 talks about doing sequencing with GENEWIZ. This should be in the "gene expression analysis based on RNA-Seq" section or a separate section describing library prep and sequencing.

- What were the parameters for sequencing in terms of read length.

- Why was the analysis done with DESeq (2010 release) instead of DESeq2 (2014 release)

- In differential expression analysis for salt stress vs no salt, it is not defined what is upregulation and downregulation. Does upregulation mean upregulated in stress samples vs non-stress samples?

What was the sequencing depth or other read stats like total reads per sample and alignment rates.

Validity of the findings

In the results, the paper characterizes the gene structure of the selected bHLH genes. Are those characterizations similar to other classes of genes in poplar?

Is the uneven chromosomal distribution independent of the size of the chromosomes?

The between-pair comparison is confusing. For example, when comparing leaf-root and stem-root, the differences identified should be the differences between leaf and stem (since root is the same dataset). However, lines 174-175 say - "gene expression are different in roots relative to both leaves and stem" which suggests something is different about root. This analysis needs to be explained better.

What is the objective of selecting the 19 genes that are deferentially expressed in the 3 sets of tissue pairs? What does it mean that these genes would be predominately deferentially expressed in both - tissue differential gene expression and differential expression due to salinity? Does it mean the genes are irrelevant for stress "tolerance" in this case?

Cluster 3 characteristics in line 182 needs to be further clarified instead of saying "opposite pattern of cluster 1".

The conclusion in line 282 about major stress responses are induced in root was also discussed in line 257-258. This seems a repetition of the result/discussion.

The paper tires to characterize a set of bHLH genes in poplar. They also try to do some functional genomic studies and validate the results using qPCR. However, the paper falls short of clearly describing the methods as well as results which makes the interpretation of the results and conclusions hard. Hence, I would recommend not accepting the paper in its current form.

---

## Round 0.2 · accepted · Accept

The manuscript has been greatly improved by addressing reviewers' concern. I suggest its acceptance.